# Crowdsource authoring as a tool for enhancing the quality of competency assessments in healthcare professions

Che-Wei Lin[1], Daniel L. Clinciu[2], Daniel Salcedo[3], Chih-Wei Huang[4], Enoch Yi No Kang[5,6,7], Yu-Chuan (Jack) Li[8,9,10,11,12] *

1 Department of Education and Humanities in Medicine, School of Medicine, College of Medicine, Taipei Medical University, Taipei, Taiwan, 2 Graduate Institute of Biomedical Science, China Medical University, Taichung, Taiwan, 3 Taipei Municipal Wanfang Hospital, Taipei, Taiwan, 4 International Center for Health Information Technology, College of Medical Science and Technology, Taipei Medical University, Taipei, Taiwan, 5 Department of Education, Wan Fang Hospital, Taipei Medical University, Taipei, Taiwan, 6 Evidence-Based Medicine Center, Wan Fang Hospital, Taipei Medical University, Taipei, Taiwan, 7 Institute of Health Policy and Management, College of Public Health, National Taiwan University, Taipei, Taiwan, 8 Graduate Institute of Biomedical Informatics, College of Medical Science and Technology, Taipei Medical University, Taipei, Taiwan, 9 International Center for Health Information Technology (ICHIT), Taipei Medical University, Taipei, Taiwan, 10 Research Center of Big Data and Meta-analysis, Wan Fang Hospital, Taipei Medical University, Taipei, Taiwan, 11 Department of Dermatology, Wan Fang Hospital, Taipei, Taiwan, 12 TMU Research Center of Cancer Translational Medicine, Taipei Medical University, Taipei, Taiwan

* jaak88@gmail.com

**Data Availability Statement:** All relevant data are within the paper and its Supporting Information files.

## Abstract

The current Objective Structured Clinical Examination (OSCE) is complex, costly, and difficult to provide high-quality assessments. This pilot study employed a focus group and debugging stage to test the Crowdsource Authoring Assessment Tool (CAAT) for the creation and sharing of assessment tools used in editing and customizing, to match specific users' needs, and to provide higher-quality checklists. Competency assessment international experts (n = 50) were asked to 1) participate in and experience the CAAT system when editing their own checklist, 2) edit a urinary catheterization checklist using CAAT, and 3) complete a Technology Acceptance Model (TAM) questionnaire consisting of 14 items to evaluate its four domains. The study occurred between October 2018 and May 2019. The median time for developing a new checklist using the CAAT was 65.76 minutes whereas the traditional method required 167.90 minutes. The CAAT system enabled quicker checklist creation and editing regardless of the experience and native language of participants. Participants also expressed the CAAT enhanced checklist development with 96% of them willing to recommend this tool to others. The use of a crowdsource authoring tool as revealed by this study has efficiently reduced the time to almost a third it would take when using the traditional method. In addition, it allows collaborations to partake on a simple platform which also promotes contributions in checklist creation, editing, and rating.

**Funding:** The funding for this study is as follows: MOST 107-2511-H-038-003; recipient: Che-Wei-Lin, MD, PhD.

**Competing interests:** The authors have declared that no competing interests exist.

## 1. Introduction

During the past 5 decades, the focus of the training of medical education has shifted dramatically towards competency-based medical education (CBME) [1]. Thus, the development of new assessment methodologies for the accurate measurement of competency in an accurate and unbiased form was needed [2]. The Objective Structured Clinical Examination (OSCE) was developed in the 1970s as a replacement for the Clinical Examination, which was a required practical competency exam for graduating physicians in the United Kingdom [3]. In the OSCE, the rater will use a checklist or rating scale to assess the student performance in a simulated environment via a standardized patient or manikin. The checklist is popular and easy to use in OSCE. OSCE has been in place for several decades, however, it attracted a significant amount of criticism, due to lack of reliability in measuring candidates' clinical competence and the effect of no implemented standardization of cases and examiners. Later developments to improve the quality of OSCE in terms of validity and reliability scores were the design of scenario and assessment tool design checklists, more frequently used as interactive digital files [4]. The basic procedure for developing OSCE checklists includes three main stages for developing assessment tools to measure clinical skills [5]. These stages constitute the traditional method of checklist development which is still used by numerous institutions around the world: 1) Preliminary List of Measurable Steps—the OSCE station authors develop an initial list of all key and measurable steps in the clinical skill to be evaluated in a particular station; 2) Specialist Assessment—experts with particular skills assess the OSCE station's draft checklists and provide feedback and suggestions for improvement; 3) Field Testing—a mock examination using realistic OSCE conditions.

Developing a new OSCE station checklist is complex, and time-consuming, it requires significant resources and the participation of experts, mock examiners, and candidates [6–9]. However, the electronic checklist system, a worldwide recent development, can eliminate missing data and decrease post-assessment working loads. However, it only provides an electronic checklist to replace the paper checklist, and therefore not helpful for faculty when designing the checklist. Thus, we hope to create an easier and more efficient system to help the faculties when designing the checklist. From the literature review, the checklist design process is complex and time-consuming. Most medical schools have the OSCE. The most used clinical skills in OSCE are the same topics. Why not make the experts collaborate to design and improve the checklist? Therefore, the quality of future medical education will rely on increased collaboration between institutions providing medical programs [10–12]. These inter-institutional collaborations could enhance the quality of medicine using joint development of educational programs and activities by combining the resources and capabilities of different medical schools [13].

One example is the MedEdPORTAL system for social media and the downloading of geographic information. It recognizes author contributions by displaying publication metrics on its website. Anyone can contribute and upload various information and data, however, only the relevant, peer-reviewed ones are accepted and saved into the system. The built-in credit system displays specific metrics such as the number of views, number of downloads, and the conversion rate from views to downloads. The system also tracks social media coverage and displays an "attention score" calculated based on these metrics. The system tracks the impact of the shared resource but indirectly acts as a reward for the authors since it is an indicator of quality and relevance.

One of the greatest examples of successful crowd authoring and crowdsourcing not only in the medical field but in all other fields is Wikipedia. This is an online encyclopedia that is completely crowd-authored; anyone with knowledge of a particular subject can contribute.

Contributions are added to the database after a peer-review process to ensure accuracy. By December 2020, Wikipedia contained more than 6 million crowdsourced articles and has become one of the largest repositories of online information in the world [14,15]. Crowdsourcing comes from a less-specific, more public group and its advantages may include improved costs, speed, quality, flexibility, scalability, or diversity. Therefore, this study aims to build a Crowdsource Assessment Authoring Tool (CAAT) which can generate an OSCE checklist easier and more efficiently. Many studies have identified the benefits of crowdsourcing, a way to gather collective wisdom effectively. However, crowdsourcing has some issues that need to be addressed. The frequently discussed ones include confidentiality, misleading, and Intellectual property rights (IPR) issues. Crowdsourcing is a practice to update or add to the information available online by soliciting the contribution of the general public to accomplish a task. All the information is public, therefore, confidentiality is a given limitation of crowdsourcing. A crowdsourced task is done by masses of people, and one of the risks is misleading. Due to a lack of knowledge or bias, the crowd might take the project in the wrong direction. Also, since all the content is from the crowd, IPR has always been a debate. Further, there is concern about collective wisdom theft since the information is publicly available [16]. Once the CAAT is built it must be tested to demonstrate usefulness and efficiency and also understand how to eliminate/resolve the possible issues mentioned.

## 2. Methods

### 2.1 CAAT system development

The online OSCE generator checklist (CAAT) conceptualized in this study was devised through two phases. The first phase established frameworks for procedure-based OSCE checklists and online platforms and it began in October 2018. Fig 1 demonstrates the CAAT (beta

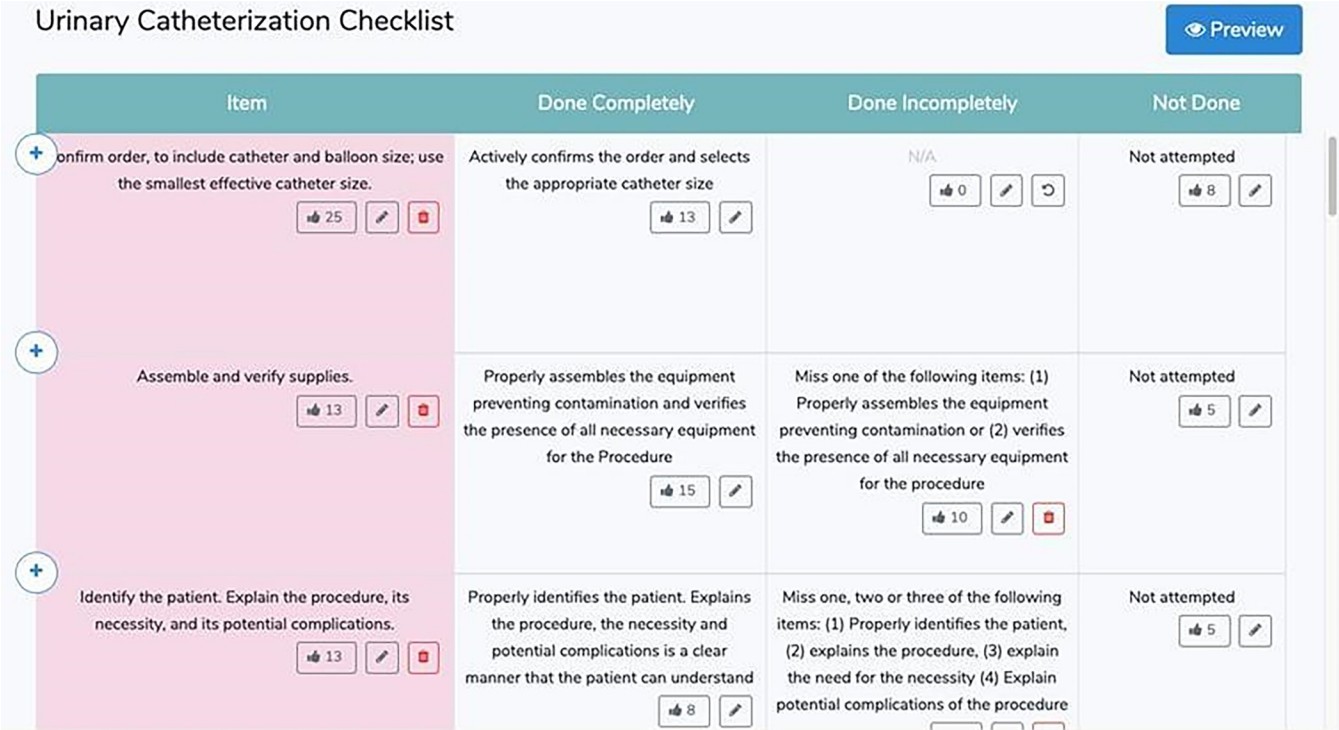

**Fig 1. The CAAT (beta 1.0) front page of the system displaying details of editable evaluation criteria in procedure-based OSCE checklists (an example of urinary catheterization).**

1.0) front page of the system displaying details of editable evaluation criteria in procedure-based OSCE checklists (an example of urinary catheterization). The second phase was aimed at detecting and fixing bugs (e.g. problems with software, and programs) and was performed in December 2018. The data collection for the study phase occurred between February and May 2019.

In Phases I and II, the focus group experts should qualify based on the following standards:

1. Medical doctors and educational scholars

2. Medical doctors must have at least three years of experience participating in the national OSCE certification or developing OSCE checklists.

3. Educational scholars must have at least three years of experience handling medical education projects or have published three medical education studies in SCI journals.

**2.1.1 Phase I: Development of the framework for the CAAT.** There are 3 assessment experts who joined the focus group to examine and adjust the OSCE's principle for drafting a CAAT framework. All are physicians (3 males, 0 females) at the TMU health system and have been involved with OSCE for over 5 years. During the drafting process, they followed the concept of ontology or mind map in order to identify the elements of the OSCE checklist and to establish their relationships. As clinical skills completion occurs in a step-by-step manner, an important element of OSCE is time series. Thus, the first layer for the CAAT is the observation of a checklist with assessment criteria and its definition. Users can edit all necessary information and make it as an individualized checklist. The second layer is a preview of the checklist to be printed and it provides the option to edit the layout accordingly. It can also be converted to an office word file. Then, the invited experts organized the first focus group framework.

During the focus group meeting, experts presented the primary concept and framework of the CAAT and guided the group in developing an OSCE checklist. According to the experience exchange and verification, the researchers further modified and enhanced the elements used in developing the OSCE checklist. Afterward, the researchers combined their knowledge and skills to extract the concepts/elements for developing an OSCE checklist. This cycle continued until no new information or modifications were produced. Then, researchers turned their confirmed findings to the engineer which implemented the CAAT's IT framework. Fig 2 demonstrates the CAAT front page of the system displaying choices of procedure and check items (an example of building a sterile system).

**2.1.2 Phase II: CAAT problems detection and fixing.** In Phase II, we invited 5 experts to review the system. Four were clinical physicians (males) and one nursing faculty (female). All have experience with designing Taiwan's medical student high stake OSCE checklists for over 5 years; one nursing faculty who was involved with the national nurse practitioner OSCE in Taiwan also participated in the Phase II development. After the engineer built the first version of the CAAT (alpha 1.0), the researchers examined the systematic problems with other invited experts. Then, the study called the second focus group to identify and fix existing problems. The same criteria for expert panels were used as in Phase I: 1) medical doctors, 2) educational scholars and 3) their required qualifications.

Five experts tested the Phase II of CAAT in a limited period (one month). The five experts used this prototype to investigate if the function and flow is appropriate. In this phase, the researchers addressed the gap between IT design and users' needs and identified some system problems. The engineer re-designed and adjusted the CAAT according to the recommendations of the panels, and the researchers re-tested the modified CAAT (alpha 1.1 version) using various medical faculty. The adjusting cycle continued until no further suggestions were given

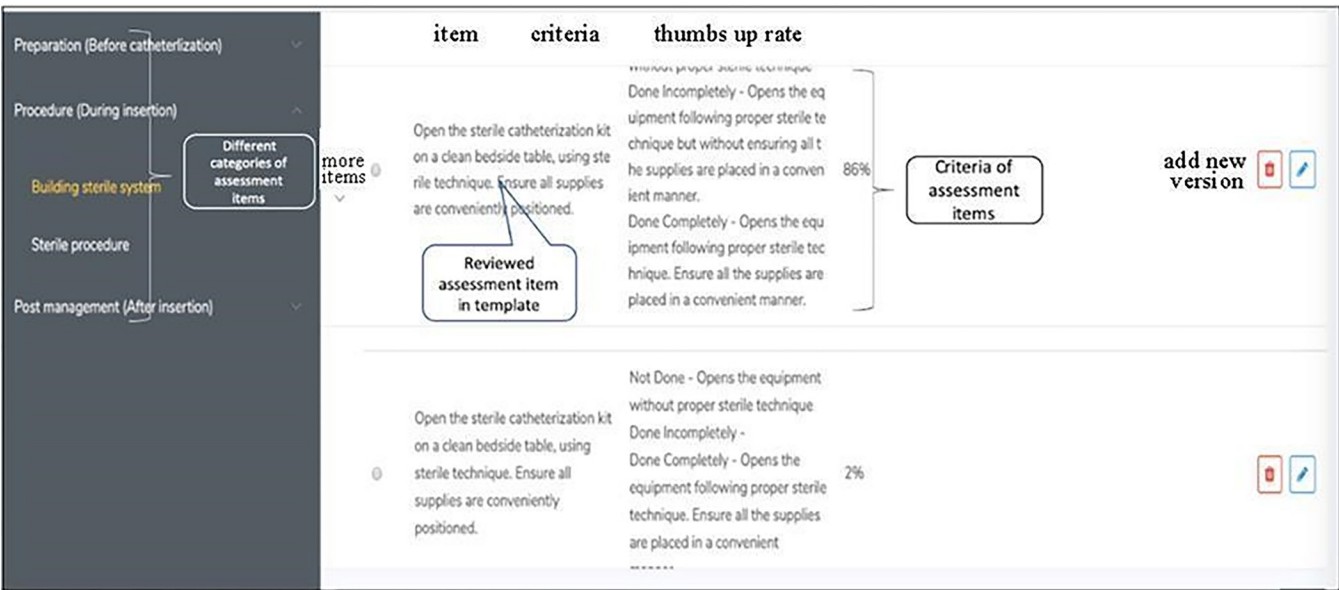

**Fig 2. The CAAT front page of the system displaying choices of procedure and check items (an example of building sterile system).**

and no problems or issues were encountered and the CAAT (beta 1.0 version, Fig 2) was approved for the study phase testing and survey.

After the above development, the CAAT (beta 1.0) system was created. Within the system, an expert can upload his checklist as a template. Other checklist designers can use the existing template and edit it accordingly to become their own checklist. The system provides a simple editing function and also can collect contributions and feedback from other crowdsource authors.

## 2.2 CAAT study phase

**2.2.1 Consent and ethics statement.** Participants were given informed consent about the study and were explained in detail how their participation can help improve collaboration, checklist creation, and to save time and resources in the healthcare profession. This study was approved by the Ethics Committee Review Board at Taipei Medical University Hospital. IRB: TMU-JIRBN201603019.

**2.2.2 The CAAT (beta version) for user test and survey.** To further verify the CAAT (beta 1.0 version) can benefit Taiwan and healthcare systems worldwide, a user test and survey for the OSCE checklist development were implemented. The initial sample size quota for this particular survey was 50 experts based on the G-power statistics ($n = 43$).

The expert participants in this study also should meet the following criteria:

1. Must be medical doctors and educational scholars

2. Medical doctors must have at least three years of experience participating in developing OSCE checklists.

3. Educational scholars must have at least three years of experience handling medical education projects

The CAAT user experience survey was performed through Google docs and included background information, the CAAT acceptance scale, and the CAAT impact. To obtain

unbiased feedback, this study used an anonymous survey format (See appendix). The CAAT acceptance scale was adopted from a verified scale [17], and consisted of 14 items reflecting four domains

of a technology acceptance model (TAM). The domains are perceived in terms of user-friendliness, usefulness, attractiveness, and intention of use. The CAAT acceptance scale involved four additional domains. The usefulness domain consists of items 1 to 4, the user-friendliness consists of items 5 to 8, attractiveness consists of items 9 to 12, and the behavioral intention consists of item 13 to item 14. Aside from the attractiveness domain, the other three domains used a 7-point Likert scale. The attractiveness domain used a 7-point bipolar scale.

## 2.3. Statistical analysis

This study used descriptive statistics (percentage event and mean with SD) for background information and inferential statistics for testing the study's hypothesis. The percentage event represented nationality, gender, and occupation. The SD mean represented experience for clinical skill teaching, checklists used for education purposes, checklist development, and the average time for developing a checklist.

For inferential statistics, an independent sample t-test, Pearson correlation, one-sample t-test, and dependent sample t-test were conducted as per hypotheses. Pearson correlation served the analysis of the correlation between the user acceptance of the CAAT and experience in clinical skill teaching (hypothesis 1), experience with checklists used for educational purposes, and experience in checklist development (hypothesis 2). A one-sample t-test was used for testing the acceptance of the CAAT among the experts (hypothesis 3). Since the questionnaire was based on a 7-point scale, the test value for the one-sample t-test was set at 5. The dependent sample t-test assessed the difference in time for developing a checklist between the CAAT and the traditional OSCE (hypothesis 4). As the p-value was lower than 0.05, the result reached statistical significance.

## 3. Results

Phases I and II are the development phases of CAAT and the CAAT study phase is to investigate the acceptance and prove the concept and are described in the methodology.

Phase I drafted the frame of the system. It produced the first layer for the CAAT which includes the observation of a checklist with assessment criteria and its definition. Users can edit all necessary information and make it an individualized checklist. The second layer is a preview of the checklist to be printed and it provides the option to edit the layout accordingly. It can also be converted to an office word file.

In Phase II, the researchers addressed the gap between IT design and users' needs and identified some system problems, and the CAAT (CAAT (beta 1.0 version, Fig 2) was approved for the study phase testing and survey.

The study phase data are as follows:

From the 60 surveys collected 50 were included in this study; 10 surveys were removed due to incomplete data. The 50 experts were from China (n = 19), Hong Kong (n = 1), Japan (n = 1), Malaysia (n = 4), Singapore (n = 1), South Korea (n = 2), Taiwan (n = 10), Thailand (n = 3), and USA (n = 9). Thirty-four (68%) were males, and 16 (32%) were females with the following professions: physicians (n = 41, 82%), and nine administrators (n = 2), bioengineers (n = 1), operation room directors (n = 1), educators (n = 2), nurses (n = 2), occupational therapists (n = 1) as seen in Table 1. The average teaching experience– 8 years; using a checklist for educational purposes– 6 years; and developing a checklist– 4.5 years.

**Table 1. Demographics and characteristics of participants.**

| Characteristics | Total No. |
|---|---|
| Males | 34 (68%) |
| Females | 16 (32%) |
| English (1st language) | 15 (30%) |
| English (2nd language) | 35 (70%) |
| Occupation | MD (41) |
| Yrs. of teaching exp.[a] | Other (9) |
| Yrs. of checklist exp.[a] | *m* |
| Developing checklists | 5–12 (8) |
| | 3–10 (6) |
| | 2–10 (4.5) |

*Note.* n = 50; *m* = mean.

## 3.1. Overall results of the CAAT acceptance survey

The scale for information system user acceptance was verified and the four domains revealed the following: The Cronbach's alpha for the usefulness domain was .90, the alpha for user friendly was .87, the alpha for attractiveness was .86, and the alpha for the behavioral intention was .87 (Table 2).

## 3.2. Participants' perception of CAAT

The following feedback was obtained from the experts participating in CAAT development and testing (Tables 2 and 3): (i) I can quickly decide the items I want to use for checking students' performance in urinary catheter insertion (M = 5.94, SD = 0.98); (ii) I can easily decide

**Table 2. The overall CAAT acceptance (test value: 5).**

| Item | MD | 95% CI Lower | Upper | *t* | *p* |
|---|---|---|---|---|---|
| 01 | .94 | .66 | 1.22 | 6.800 | < .001 |
| 02 | 1.00 | .76 | 1.24 | 8.250 | < .001 |
| 03 | .92 | .65 | 1.19 | 6.890 | < .001 |
| 04 | .84 | .51 | 1.17 | 5.090 | < .001 |
| 05 | 1.06 | .81 | 1.31 | 8.421 | < .001 |
| 06 | 1.22 | .96 | 1.48 | 9.480 | < .001 |
| 07 | 1.16 | .95 | 1.37 | 11.108 | < .001 |
| 08 | 1.02 | .74 | 1.30 | 7.366 | < .001 |
| 09 | 1.02 | .77 | 1.27 | 8.087 | < .001 |
| 10 | .82 | .51 | 1.13 | 5.358 | < .001 |
| 11 | .80 | .52 | 1.08 | 5.838 | < .001 |
| 12 | .76 | .48 | 1.04 | 5.367 | < .001 |
| 13 | 1.08 | .79 | 1.37 | 7.584 | < .001 |
| 14 | .88 | .53 | 1.23 | 5.088 | < .001 |
| Useful | .93 | .70 | 1.15 | 8.203 | < .001 |
| Ease | 1.12 | .91 | 1.32 | 10.792 | < .001 |
| Enjoy | .85 | .59 | 1.11 | 6.611 | < .001 |
| BI | .98 | .69 | 1.27 | 6.861 | < .001 |

One-sample *t*-test; BI, behavior intention; CI, confidence interval; MD., mean difference.

**Table 3. Difference in developing a checklist between the CAAT and traditional method.**

|  | CAAT | | Traditional | | | | Cohen's d | p-value |
|---|---|---|---|---|---|---|---|---|
| **Group** | **M** | **SD** | **M** | **SD** | **t** | **d** | **power** | **p** |
| Overall | 65.76 | 205.92 | 167.9 | 501.25 | -2.36 | -0.61 | 0.68 | 0.022 |
| **Languages** | | | | | | | | |
| English | 25.2 | 20.25 | 54.67 | 36.57 | -2.98 | -1.59 | 0.88 | 0.01 |
| Non-English | 83.14 | 244.75 | 216.43 | 594.54 | -2.18 | -0.75 | 0.69 | 0.036 |
| **Experience for checklist development** | | | | | | | | |
| Junior | 41.12 | 70.8 | 96.6 | 75.44 | -3.76 | -1.54 | 0.98 | 0.001 |
| Senior | 90.4 | 283.36 | 239.2 | 704.77 | -1.75 | -0.71 | 0.52 | 0.093 |

Paired *t*-test. M, mean; SD, standard deviation. [a] English speaker (n = 15) and non-English speaker (n = 35); [b] senior 7 (experience on developing checklist more than 4 years, n = 25) and junior (experience on developing checklist within 8 4 years, n = 25).

the items. I want to use for checking students' performance in urinary catheter insertion (M = 6.00, SD = 0.86); (iii) I can develop an accurate checklist to assess students' performance in urinary catheter insertion (M = 5.92, SD = 0.94); (iv) the CAAT provides me with information that allows me to develop an effective checklist for urinary catheter insertion (M = 5.84, SD = 1.17); (v) the interaction with CAAT is clear and understandable (M = 6.06, SD = 0.89); (vi) the interaction with CAAT does not increase the workload when creating a checklist (M = 6.22, SD = 0.91); (vii) I find CAAT easy to use (M = 6.16, SD = 0.74); (viii) it is easy to get CAAT to do what I need (M = 6.02, SD = 0.98); (ix) I feel CAAT is enjoyable/awful (M = 6.02, SD = 0.89); (x) I feel CAAT is exciting/dull (M = 5.82, SD = 1.08); (xi) I feel CAAT is pleasant/unpleasant (M = 5.80, SD = 0.97); (xii) I feel CAAT is interesting/boring (M = 5.76, SD = 1.00); (xiii) I intend to revisit CAAT in the future (M = 6.08, SD = 1.01); and (xiv) I will use CAAT next time I need to generate OSCE checklist (M = 5.88, SD = 1.22).

The results of the one-sample t-test with a test value of 5 showed that the experts CAAT to user (MD = 0.80, $t = 5.84$, $p < .001$); (xii) how interesting is CAAT (MD = 0.76, $t = 5.37$, $p < .001$); (xiii) user intends to use CAAT in the future (MD = 1.08, $t = 7.58$, $p < .001$); and (xiv) user will likely use CAAT for generating the next OSCE checklist (MD = 0.88, $t = 5.09$, $p < .001$).

The experts agreed that CAAT is a useful, easy, and interesting system for generating a checklist for educational purposes (Table 2). More specifically, the results showed that: (i) a user can quickly decide which items to use for checking students' performance in urinary catheter insertion (MD = 0.94, $t = 6.80$, $p < .001$); (ii) can easily decide which item to use for checking students' performance in urinary catheter insertion (MD = 1.00, t = 8.25, p < .001); (iii) can develop an accurate checklist to assess students' performance in urinary catheter insertion (MD = 0.92, t = 6.89, p < .001); (iv) CAAT provides information in ways to effectively develop a checklist for urinary catheter insertion (MD = 0.84, t = 5.09, p < .001); (v) interaction with CAAT is efficient and responsive (MD = 1.06, t = 8.42, p < .001); (vi) CAAT does not increase workload when creating a checklist (MD = 1.22, t = 9.48, p < .001); (vii) CAAT is easy to use (MD = 1.16, t = 11.12, p < .001); (viii) CAAT is adjusted to user's needs (MD = 1.02, t = 7.37, p < .001); (ix) the degree of like/dislike (MD = 1.02, t = 8.09, p < .001); (x) CAAT is exciting to use (MD = 0.82, t = 5.36, p < .001); (xi) its user appeal.

Overall, experts around the world that participated indicate that the CAAT is a useful (MD = 0.93, $t = 8.20$, $p < .001$), easy to use (MD = 1.12, $t = 10.79$, $p < .001$), and enjoyable (MD = 0.85, $t = 6.61$, $p < .001$) platform for generating a checklist for clinical skill. Moreover, they expressed keen interest in using the CAAT in future (MD = 0.98, $t = 6.86$, $p < .001$).

### 3.3. Effectiveness of the CAAT

Overall, the experts used 65.76 minutes for developing a new checklist using CAAT, whereas, they spent 167.90 minutes building a new checklist traditionally (Table 3). The dependent-sample $t$-test revealed that CAAT saved 102 minutes in generating a new checklist as compared to the traditional method ($t$ = -2.36, $p < .05$).

In stratified analyses, this study explored the effectiveness of the CAAT among English native speakers and non-native speakers respectively; the study also separated data for understanding the effectiveness of the CAAT among junior experts and senior experts in developing checklists. The English native speakers generated a new checklist in about 25.20 minutes using CAAT, and in 54.67 minutes traditionally. The mean difference between the two approaches was -29.47 minutes ($t$ = -2.98, $p < .05$). The non-native speakers generated a new checklist using CAAT in about 83.14 minutes, and in 216.43 minutes traditionally. The mean difference between the two approaches was -133.29 minutes ($t$ = -2.18, $p < .05$). When the study separated junior experts from senior experts, the results demonstrated that the junior experts generated a new checklist with CAAT in about 41.12 minutes, and with the traditional way in 96.60 minutes. The mean difference between the two approaches was -55.48 minutes ($t$ = -3.76, $p = .001$). However, among the senior experts, the results demonstrated that they generated a new checklist with CAAT in about 90.40 minutes, and with the traditional method in 239.20 minutes. The mean difference between the two approaches was -148.80 minutes with marginal significance ($t$ = -1.75, $p = .093$). Participants/experts from around the world perceived that CAAT could change current practices of checklist development (item 12: MD = 0.76, t = 5.07, p < .001; Table 2).

## 4. Discussion

This pilot study developed and tested the Crowdsource Authoring Assessment Tool (CAAT), a new software and collaboration system to faster and more efficiently generate a checklist for assessing the clinical skill performance of learners. Analyzed data collected from 50 recognized international experts in the field of competency assessment revealed that CAAT can significantly reduce time and improve efficiency and ease when generating a checklist to assess a learner's clinical skills (Table 3).

The average time for participating experts was 65.76 minutes when using CAAT, whereas, they spent 167.90 minutes using OSCE, the traditional method currently used worldwide [1–3]. According to the study's results, the average time saved by using CAAT is 102.14 minutes (Table 3). It showed that the CAAT system really saves the time of checklist design. The template function can guide the checklist designer to the basic and important assessment criteria of a particular skill. Moreover, the editing function also provides an individualized checklist for particular skills. The CAAT provides an efficient and innovative way to design the checklist.

In addition, Cronbach's alpha revealed high reliability for usefulness (.90), user-friendliness (.87), attractiveness (.86), and behavioral intention (.87) are all important features of CAAT (Table 2). To analyze these important features of CAAT in an unbiased and comprehensive manner, experts from around the world were invited to participate, which are both native (30%) and non-native English speakers (70%). In addition, the experts participating in this study comprised a variety of professionals from physicians to bioengineers (Table 1). It is important to have a diverse group of experts from around the world in order to better assess the efficiency of the tool in terms of its usefulness, ease of use and from a non-English native perspective [10,11].

The study provided certain items for participants to use to determine if they can faster and more accurately create a checklist in assessing a test taker's performance (e.g. the insertion of a

urinary catheter). In the CAAT acceptance survey, value 4 is intermediate because of the 7-point scale used. Thus, a test value higher than 5 means a positive review and the items of CAAT assessed by experts received a positive perception overall (Table 2).

CAAT adapted some of Wikipedia's while also verifying the user's identity. The user's instructor identity needs to be confirmed before being allowed into the system. The user is also required to sign the NDA (non-disclosure agreement). These mechanisms will be able to overcome some of the shortcomings of crowdsourcing. The content on CAAT is all about assessment checklists. Therefore, it is important to ensure that students have no access to the content. During the onboarding process, the user's competence in checklist design will reveal reducing the possibility of editing errors. Further, there are checklist templates available on CAAT. Templates are created and provided by skilled experts. Users can edit based on the template to address the specific needs of their institutions. The system allows users to edit within a direction range to reduce errors due to bias or lack of knowledge. When experts are solicited to provide templates, they need to sign an IPR (intellectual property rights) release to the system. At login, users are reminded that their editing will be logged by the system, and the IPR belongs to CAAT. The NDA ensures that the checklists will not leak. These are the mechanisms to help CAAT enjoy the benefits of crowdsourcing and avoid its downsides [18].

A potential limitation of crowdsourcing is the development of localized assessment checklists that might not be compatible with other settings. Some settings may have specific procedures or equipment for a particular skill, that are not generalizable. Even though in our design we have already tried to provide an editing function it's difficult to create checklists that are a good fit with all settings. In the crowdsourcing methodology, the quality of the template is a critical issue. The user will modify the template and create a checklist to fit their particular settings. If the quality of the template is inadequate the users may spend a significant amount of time editing and improving the content. It is necessary to implement some mechanism to ensure the content of the template meets certain quality metrics. Currently, we use experts to review the uploaded checklist template. In the future, Natural Language Processing (NLP) based Artificial Intelligence (AI) may help determine if templates meet quality standards and suggest modifications to improve checklist generalizability. In future developments, we will encourage the OSCE experts to upload more checklist templates. We will provide the checklist bank to the scholars who will design the checklists. The more faculties use and edit the checklists the more accurate and efficient the template will be. Moreover, extended use of AI in the future will help to review the templates.

Another limitation of our study is its small sample size. We only invited experts who participated in assessment training courses throughout the four popular simulation centers worldwide. Moreover, the experts needed to spend an additional 30 min for a real experience with CAAT prior to the survey. Even with additional announcements and promotion of CAAT, the response rate of the experts interested to participate in this study was low. The language barrier could be another limitation as two-thirds of the participants were from non-English speaking countries. The CAAT is still in its development stage and it was mostly restricted to OSCE checklists. In the future, it can be expanded to a point-of-care assessment of competency, self, and patient-based assessments.

## Supporting information

**S1 Fig. The CAAT version 1.0 editing window.**
(DOCX)

**S2 Fig. The CAAT display of a newly added assessment item.**
(DOCX)

**S3 Fig. The displaying of a completed assessment on the CAAT system.**
(DOCX)

**S4 Fig. The CAAT interface displaying a deleted assessment criteria.**
(DOCX)

**S5 Fig. CAAT's Information system framework.**
(DOCX)

**S1 Table. Correlations between experience and the CAAT acceptance.**
(DOCX)

**S2 Table. Differences in the CAAT acceptance between senior and junior experts.**
(DOCX)

**S1 Data.**
(XLSX)

## Acknowledgments

The authors would like to thank all the participants and all those involved in helping out with the crowdsourcing tool development and assessment.

## Author Contributions

**Conceptualization:** Che-Wei Lin.

**Data curation:** Che-Wei Lin, Daniel Salcedo, Chih-Wei Huang.

**Formal analysis:** Che-Wei Lin, Daniel L. Clinciu, Yu-Chuan (Jack) Li.

**Investigation:** Chih-Wei Huang, Enoch Yi No Kang.

**Methodology:** Che-Wei Lin, Enoch Yi No Kang.

**Project administration:** Yu-Chuan (Jack) Li.

**Resources:** Chih-Wei Huang.

**Software:** Enoch Yi No Kang.

**Supervision:** Chih-Wei Huang, Yu-Chuan (Jack) Li.

**Visualization:** Daniel Salcedo.

**Writing – original draft:** Che-Wei Lin, Daniel L. Clinciu.

**Writing – review & editing:** Daniel L. Clinciu, Daniel Salcedo.

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
