## [Decision Letter · Decision Letter 0]

28 Jan 2022

PONE-D-21-35425Crowdsource Authoring as a Tool for Enhancing the Quality of Competency Assessments in Healthcare ProfessionsPLOS ONE

Dear Authors,

Thank you for submitting your manuscript to PLOS ONE. After careful consideration, we feel that it has merit but does not fully meet PLOS ONE’s publication criteria as it currently stands. Therefore, we invite you to submit a revised version of the manuscript that addresses the points raised during the review process.

ACADEMIC EDITOR: Please see comments below.

We look forward to receiving your revised manuscript.

Kind regards,

Dejan Dragan, PhD

Academic Editor

PLOS ONE

Journal Requirements:

2. Please provide additional details regarding participant consent. In the ethics statement in the Methods and online submission information, please ensure that you have specified what type you obtained (for instance, written or verbal, and if verbal, how it was documented and witnessed). If your study included minors, state whether you obtained consent from parents or guardians. If the need for consent was waived by the ethics committee, please include this information

3. We note that you have stated that you will provide repository information for your data at acceptance. Should your manuscript be accepted for publication, we will hold it until you provide the relevant accession numbers or DOIs necessary to access your data. If you wish to make changes to your Data Availability statement, please describe these changes in your cover letter and we will update your Data Availability statement to reflect the information you provide

Additional Editor Comments (if provided):

The reviewers have completed their review. In general, they are fairly satisfied with the manuscript. Nevertheless, there are some minor issues that are needed to be corrected. Accordingly, the AE's decision is: A minor revision. AE DD.

Reviewers' comments:

Reviewer's Responses to Questions

**Comments to the Author**

1. Is the manuscript technically sound, and do the data support the conclusions?

Reviewer #1: Yes

Reviewer #2: Yes

2. Has the statistical analysis been performed appropriately and rigorously? 

Reviewer #1: Yes

Reviewer #2: Yes

3. Have the authors made all data underlying the findings in their manuscript fully available?

Reviewer #1: No

Reviewer #2: Yes

4. Is the manuscript presented in an intelligible fashion and written in standard English?

Reviewer #1: Yes

Reviewer #2: No

5. Review Comments to the Author

Reviewer #1: this is a valuable work to be published but needs some revisions as followings:

1) Provide the demographic characteristics of phases 1 and 2 participants.

2) Authors should report the results for each phase of study separately. So, they should re-organize the results section and provide the findings of all study phases.

3) In discussion, authors should compare the performance of designed tool with the same ones to highlight the strengths of their work.

Reviewer #2: Thank you for the opportunity to review this interesting manuscript that describes and evaluates a crowd source approach to developing competency assessment tools for health care professionals. The new tool seeks to replace the OSCE which has been in use since the 1970s and it is good to see a novel approach to the design of competency assessment. The authors used a three step approach to first design the tool, build and refine it before testing it with relevant professional and educators.

The manuscript was generally very positive in tone and could at times have been more critical for example, what are some of the disadvantages of Wikipedia and how could these be address with this new methodology?

The methods are appropriate and well described, though I was unable to see when the study took place. The results are clearly laid out.

In the discussion it would be interesting to hear more about how the tool could be developed.

There are some minor grammatical amendments (some noted in the attached) that would improve the reading of this manuscript.

6. PLOS authors have the option to publish the peer review history of their article (what does this mean?). If published, this will include your full peer review and any attached files.

Reviewer #1: **Yes: **Mohammad Amin Bahrami

Reviewer #2: No

---

## [Author Response · Author response to Decision Letter 0]

26 Apr 2022

Reviewer 1 Provide the demographic characteristics of phases 1 and 2 participants.

 Response: The demographics for Phase I and II participants are found on lines 97, 116 – 119 (red highlights. For the study participants, the demographics are found in Table 1. 

 Authors should report the results for each phase of study separately. So, they should re-organize the results section and provide the findings of all study phases. 

Response: We reported Phase I, II and the Study Phase results on lines 182 to 201 (please see the red highlights)

 In discussion, authors should compare the performance of designed tool with the same ones to highlight the strengths of their work. 

Response: The comparisons are found on lines 256 – 279: The average time for participating experts was 65.76 minutes when using CAAT, whereas, they spent 167.90 minutes using OSCE, the traditional method currently used worldwide [1-3]. According to the study’s results, the average time saved by using CAAT is 102.14 minutes (Table 3). It showed that the CAAT system really saves the time of checklist design. The template function can guide the checklist designer the basic and important assessment criteria of particular skill. Moreover, the editing function also provides the individualized checklist for particular skills. The CAAT provides an efficient and innovative way to design the checklist. 

Reviewer 2 I was unable to see when the study took place. 

Response: The study occurred between October 2018 and May 2019 (we rewrote it in the abstract). It is also found in the methods section lines 83 and 86 (red highlights).

 Response: In the discussion it would be interesting to hear more about how the tool could be developed. In the discussion we have added more about the tool on lines 284-294:

In the crowdsourcing methodology, the quality of the template is a critical issue. The user will modify the template and create a checklist to fit their particular settings. If the quality of the template is inadequate the users may spend a significant amount of time editing and improving the content. It is necessary to implement some mechanism to ensure the content of the template meets certain quality metrics. Currently, we use experts to review the uploaded checklist template. In the future, Natural Language Processing (NLP) based Artificial Intelligence (AI) may help determine if templates meet quality standards and suggest modifications to improve checklist generalizability.

In future developments, we will encourage the OSCE experts to upload more checklist templates. We will provide the checklist bank to the scholars who will design the checklists. The more faculties use and edit the checklists the more accurate and efficient the template will be. Moreover, extended use AI in the future will help to review the templates.

---

## [Decision Letter · Decision Letter 1]

21 Jul 2022

PONE-D-21-35425R1Crowdsource Authoring as a Tool for Enhancing the Quality of Competency Assessments in Healthcare ProfessionsPLOS ONE

Dear Dr. Li,

Thank you for submitting your manuscript to PLOS ONE. After careful consideration, we feel that it has merit but does not fully meet PLOS ONE’s publication criteria as it currently stands. Therefore, we invite you to submit a revised version of the manuscript that addresses the points raised during the review process.

 Your manuscript has been reassessed by the two reviewers from the previous round, whose reports can be found below. As you will see from the comments, the reviewers acknowledge that the manuscript has improved, but there remain some points raised by reviewer 2 where it is not obvious if they have been addressed. Please update your response to reviewers document to include an entry corresponding to each of the highlights/comments made on the pdf document submitted by reviewer 2 (reattached here for convenience), and ensure that the changes you made in response to their comments are clearly highlighted in the tracked changes version of the manuscript, so that the reviewer can easily assess how the manuscript has been revised.

We look forward to receiving your revised manuscript.

Kind regards,

Joseph Donlan

Editorial Office

PLOS ONE

Journal Requirements:

Reviewers' comments:

Reviewer's Responses to Questions

**Comments to the Author**

1. If the authors have adequately addressed your comments raised in a previous round of review and you feel that this manuscript is now acceptable for publication, you may indicate that here to bypass the “Comments to the Author” section, enter your conflict of interest statement in the “Confidential to Editor” section, and submit your "Accept" recommendation.

Reviewer #1: All comments have been addressed

Reviewer #2: (No Response)

2. Is the manuscript technically sound, and do the data support the conclusions?

Reviewer #1: Yes

Reviewer #2: Yes

3. Has the statistical analysis been performed appropriately and rigorously? 

Reviewer #1: Yes

Reviewer #2: N/A

4. Have the authors made all data underlying the findings in their manuscript fully available?

Reviewer #1: Yes

Reviewer #2: Yes

5. Is the manuscript presented in an intelligible fashion and written in standard English?

Reviewer #1: Yes

Reviewer #2: Yes

6. Review Comments to the Author

Reviewer #1: authors have addressed all of my requested comments, with thanks to the authors i have no more comments

Reviewer #2: Thank you for the revisions undertaken to date. Unfortunately I was unable to identify them in the manuscript - there were no track changes or highlighted sections in the manuscript I reviewed.

In the response to reviewers I can see two comments have been responded to. However one was overlooked. From my initial review..."The manuscript was generally very positive in tone and could at times have been more critical for example, what are some of the disadvantages of Wikipedia and how could these be address with this new methodology?" Could the introduction be reviewed with a more critical eye?

7. PLOS authors have the option to publish the peer review history of their article (what does this mean?). If published, this will include your full peer review and any attached files.

Reviewer #1: No

Reviewer #2: No

---

## [Author Response · Author response to Decision Letter 1]

3 Oct 2022

we have addressed the concerns of reviewer 2 and have provided the files according to the editor.

---

## [Decision Letter · Decision Letter 2]

21 Nov 2022

Crowdsource Authoring as a Tool for Enhancing the Quality of Competency Assessments in Healthcare Professions

PONE-D-21-35425R2

Dear colleagues, 

We’re pleased to inform you that your manuscript has been judged scientifically suitable for publication and will be formally accepted for publication once it meets all outstanding technical requirements.

Kind regards,

Yaser Mohammed Al-Worafi

Academic Editor

PLOS ONE

Reviewers' comments:

Reviewer's Responses to Questions

**Comments to the Author**

1. If the authors have adequately addressed your comments raised in a previous round of review and you feel that this manuscript is now acceptable for publication, you may indicate that here to bypass the “Comments to the Author” section, enter your conflict of interest statement in the “Confidential to Editor” section, and submit your "Accept" recommendation.

Reviewer #1: All comments have been addressed

Reviewer #3: All comments have been addressed

2. Is the manuscript technically sound, and do the data support the conclusions?

Reviewer #1: Yes

Reviewer #3: Yes

3. Has the statistical analysis been performed appropriately and rigorously? 

Reviewer #1: Yes

Reviewer #3: Yes

4. Have the authors made all data underlying the findings in their manuscript fully available?

Reviewer #1: Yes

Reviewer #3: Yes

5. Is the manuscript presented in an intelligible fashion and written in standard English?

Reviewer #1: Yes

Reviewer #3: Yes

6. Review Comments to the Author

Reviewer #1: thanks to the authors

All of my comments have been addressed, so i have no more comments. the manuscript is acceptable in my opinion

Reviewer #3: The comments of the previous reviewers have been addressed.

The authors of study proposes the idea of crowdsource authoring tool to replace the OSCE, and performs pilot run and a phase 2 trial to find out and rectify any issues with the pilot, and also assess the acceptability of the CAAT which is appreciated.

The structure of the manuscript follows the author guidelines.

7. PLOS authors have the option to publish the peer review history of their article (what does this mean?). If published, this will include your full peer review and any attached files.

Reviewer #1: **Yes: **Dr. Mohammad Amin Bahrami

Reviewer #3: No

---

## [Editor Report · Acceptance letter]

13 Mar 2023

PONE-D-21-35425R2 

Crowdsource Authoring as a Tool for Enhancing the Quality of Competency Assessments in Healthcare Professions 

Dear Dr. Li:

I'm pleased to inform you that your manuscript has been deemed suitable for publication in PLOS ONE. Congratulations! Your manuscript is now with our production department. 

Kind regards, 

on behalf of

Professor Yaser Mohammed Al-Worafi 

Academic Editor

PLOS ONE